**Correspondence**

# The dual luciferase assay does not support claims of stop codon readthrough on the *AGO1* mRNA

Apoorva N Suresh, John F Atkins ⬤ & Gary Loughran ⬤ ✉

Comment on: A Singh et al
See reply: A Singh et al
See also: LE Manjunath et al

In "Let-7a-regulated translational readthrough of mammalian *AGO1* generates a microRNA pathway inhibitor" by Singh et al, 2019, and "Transcript-specific induction of stop codon readthrough using a CRISPR-dCas13 system" by Manjunath et al, 2024, the authors propose that stop codon readthrough of *AGO1* mRNA generates a C-terminally extended AGO1 proteoform that inhibits the miRNA pathway. It is reported that ~20% of the ribosomes translating the *AGO1* mRNA do not recognize the annotated stop codon and instead insert an amino acid from a near-cognate tRNA—a well-studied phenomenon termed stop codon readthrough. As far as we know, this is the highest reported readthrough efficiency observed for any human mRNA. In general, stop codon readthrough is an inefficient process (<1%), although several occurrences of efficient readthrough (up to 17%) on human mRNAs have been described where well-defined nucleotide contexts are required (Loughran et al, 2014, 2017, 2018).

Readthrough efficiency is often measured with dual reporter systems where the test stop codon and its surrounding nucleotides are flanked by an upstream reporter (often Renilla luciferase; R-luc) and a downstream reporter (often Firefly luciferase; F-luc). Both reporters are in the same reading frame and expressed from a single mRNA. Ribosomes that readthrough the stop codon synthesize an R-luc-F-luc fusion protein, whereas those that terminate at the stop codon yield R-luc alone (Grentzmann et al,

1998). Readthrough efficiency is calculated as the F-luc/R-luc activities of the test construct relative to the F-luc/R-luc activities from a sense codon control reporter. Previously, we generated an updated dual-luciferase reporter system (pSGDLuc) that reduces potential artefacts associated with altered reporter activity or stability that can occur when reporters are expressed as long fusion proteins (Loughran et al, 2017). This dual reporter vector insulates the test sequence from the reporter sequence by the introduction of flanking StopGo/2A elements derived from foot and mouth disease virus. StopGo/2A elements allow co-translational hydrolysis of a specific peptidyl-tRNA linkage without disrupting continued ribosome elongation. Therefore, reporter proteins are co-translationally separated from the test sequence product and have the same amino acid sequence irrespective of the test sequence.

To estimate the readthrough efficiency of the *AGO1* stop codon, these authors used a dual-luciferase reporter system in which AGO1-F-luc and R-luc were translated from separate mRNAs. AGO1-F-luc included ~800 nucleotides of the *AGO1* mRNA flanking its annotated stop codon (~700 upstream and ~100 downstream) plus a linker sequence fused to the same reading frame as F-luc. Similar to the dual-luciferase reporters expressed from a single mRNA described above, readthrough efficiency is calculated as the F-luc/R-luc activities of the test construct relative to the F-luc/R-luc activities from a sense codon control reporter.

Because of our long-standing interest in stop codon readthrough, we were curious to explore the mechanism responsible for remarkably efficient readthrough on the

*AGO1* mRNA. When we tested the *AGO1* stop codon and its surrounding nucleotides using our single mRNA StopGo/2A dual-luciferase system (pSGDLuc), we could not detect readthrough activity above 0.1% (Fig. 1A,B). In agreement with previous reports (Loughran et al, 2014, 2017; De Bellis et al, 2017), efficient stop codon readthrough (~6%) was observed on *AQP4*. Next, we generated the same AGO1-F-luc reporters used by these authors. To control for possible differences in transfection efficiencies with separate plasmids (one each for F-luc and R-luc), we expressed both reporters from the same plasmid using a bidirectional promoter. Concerned by possible negative impacts on F-luc activity or stability resulting from the fusion of 232 AGO1 amino acids to its N-terminus, we also generated unfused versions of the AGO1-F-luc constructs by introducing a StopGo/2A element immediately before F-luc (AGO1-SG-F-luc). From the fused AGO1-F-Luc reporters, we detected apparent readthrough, albeit at lower efficiency (~10%) than previously reported (Singh et al, 2019) (Fig. 1C). Unlike Singh et al, we did not observe significant differences in apparent readthrough efficiencies between different stop codons. However, when we tested the unfused AGO1-SG-F-luc constructs, apparent readthrough almost disappeared (Fig. 1C). To investigate this discrepancy, we first compared the absolute luciferase activities of the sense controls between fused and unfused reporters. Although there were no significant differences in R-luc activities, F-luc values were approximately eightfold lower in the absence of StopGo (fused), indicating that fusion of 232 AGO1 amino acids to F-luc reduced its activity or stability (Fig. 1D). However, absolute F-luc activities from all stop codon

School of Biochemistry and Cell Biology, University College Cork, Cork, Ireland. ✉E-mail: g.loughran@ucc.ie
https://doi.org/10.1038/s44318-025-00478-1 | Published online: 11 June 2025

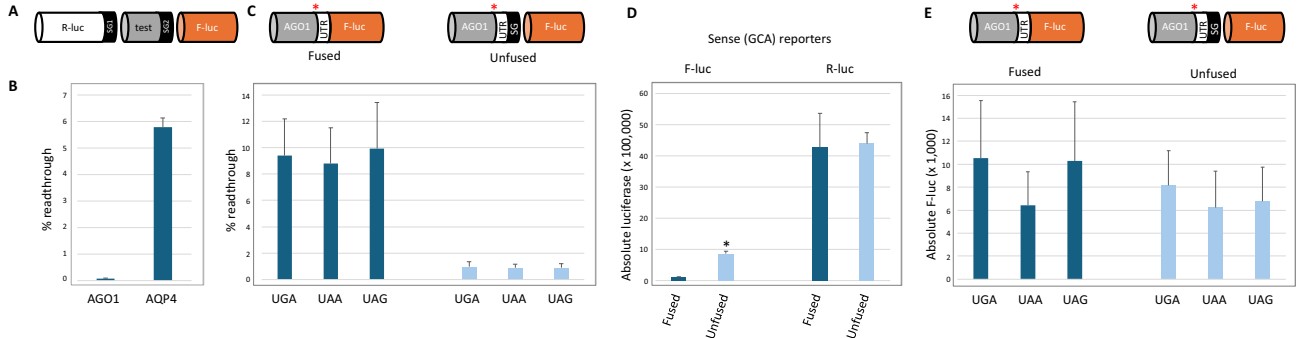

**Figure 1. Stop codon readthrough analysis of *AGO1* by different dual-luciferase reporter systems.**

(A) Illustration of the reporter proteins expressed from unfused dual-luciferase reporter systems (pSGDLuc). (B) Readthrough efficiencies (%) of *AGO1* and *AQP4* stop codons calculated by dual-luciferase assay in HEK293T cells. SG refers to the StopGo element from foot and mouth disease virus. (C) Readthrough efficiencies (%) from AGO1 fused and unfused reporters calculated by dual-luciferase assay in HEK293T cells. (D) Absolute luciferase activities (F-luc and R-luc) from sense codon fused and unfused reporters (*$P < 10^{-6}$). (E) Absolute F-luc activities from AGO1 fused and unfused stop codon reporters. SG refers to StopGo. Red asterisk indicates position of test stop codon or control sense codon. $N = 3$. Source data are available online for this figure.

**Table 1. Oligonucleotides and gene block used in this study.**

| Primer 1 | AGO1 RT UGA S | ATAACTCGAGCTTCGCTTGAAGGCAGAACGCTGTTACCTCACTGGATAGAAGAAAGC |
|---|---|---|
| Primer 2 | AGO1 RT GCA S | ATAACTCGAGCTTCGCTGCAAGGCAGAACGCTGTTACCTCACTGGATAGAAGAAAGC |
| Primer 3 | AGO1 RT AS | TTATAGATCTCAGCTCCTGGGGCTTGGAAAGCTTTCTTCTATCCAGTG |
| Primer 4 | AGO1 RT S | AGGCAGAACGCTGTTACCTC |
| Primer 5 | AGO1 RT GCA AS | GAGGTAACAGCGTTCTGCCTTGCAGCGAAGTACATGGTGCGCAGAGTATCC |
| Primer 6 | AGO1 RT UAA AS | GAGGTAACAGCGTTCTGCCTTTAAGCGAAGTACATGGTGCGCAGAGTATCC |
| Primer 7 | AGO1 RT UAG AS | GAGGTAACAGCGTTCTGCCTCTAAGCGAAGTACATGGTGCGCAGAGTATCC |
| gBlock | AGO1 RT UGA | TCACTATAGGCTAGCCACCATGACTGTGCGGGTACAGCGACCACGGCAAGAGATCATTGAAGACTTGTCCTACATGGTGCGTGAGCTCCTCATCCAATTCTACAAGTCCACCCGTTTCAAGCCTACCCGCATCATCTTCTACCGAGATGGGGTGCCTGAAGGCCAGCTACCCCAGATACTCCACTATGAGCTACTGGCCATTCGTGATGCCTGCATCAAACTGGAAAAGGACTACCAGCCTGGGATCACTTATATTGTGGTGCAGAAACGCCATCCACACCCGCCTTTTCTGTGCTGACAAGAATGAGCGAATTGGGAAGAGTGGTAACATCCCAGCTGGGACCACAGTGGACACCAACATCACCCACCCATTTGAGTTTGACTTCTATCTGTGCAGCCACGCAGGCATCCAGGGCACCAGCCGACCATCCCATTACTATGTTCTTTGGGATGACAACCGTTTCACAGCAGATGAGCTCCAGATCCTGACGTACCAGCTGTGCCACACTTACGTACGATGCACACGCTCTGTCTCTATCCCAGCACCTGCCTACTATGCCCGCCTGGTGGCTTTCCGGGCACGATACCACCTGGTGGACAAGGAGCATGACAGTGGAGAGGGGAGCCACATATCGGGGCAGAGCAATGGGCGGGACCCCCAGGCCCTGGCCAAAGCCGTGCAGGTTCACCAGGATACTCTGCGCACCATGTACTTCGCTTGAAGGCAGAACGCTGTTACCTCACTGGATAGAAGAAAGCTTTCCAAGCCCCAGGAGCTGTGCCACCCAAATCCAGAGGAAGCAAGGAGGAGGGAGGTGGGGGGCGGCTCCGGCGGCTCCCTCGTGCTCGGGGGGATCCATAAAT |

(test) reporters did not differ significantly, regardless of the presence of StopGo (Fig. 1E), suggesting a basal level of intrinsic F-luc activity from these reporters independent of readthrough (otherwise we would expect the same approximately eightfold difference in F-luc values between fused and unfused stop codon (test) reporters that we observed for the sense codon reporters). Low levels of reporter activity from plasmid-based systems are not uncommon (Khan et al, 2022; Loughran et al, 2022, 2025). This activity is usually derived from unexpected mRNAs generated by either cryptic splicing or cryptic promoters. We believe that the low levels of F-luc activity observed here for all stop codon reporters result from such artefacts. This, coupled with the compromised activity of the fused sense control, resulted in an overestimation of AGO1 readthrough since readthrough efficiencies are calculated relative to the sense control.

## Methods

### Expression constructs

AGO1 single mRNA dual-luciferase constructs were generated by overlapping PCR with primers 1–3 indicated in Table 1, digested with *Xho*I / *Bgl*II and ligated into *Psp*XI / *Bgl*II digested pSGDlucV3.0 (Addgene 119760). AQP4 reporters were generated previously (Loughran et al, 2017). AGO1 separate mRNA dual-luciferase construct (UGA) was generated by first cloning gBlock (Table 1) *Nhe*I / *Bam*HI into p2Luc (Grentzmann et al, 1998) to generate p2Luc-AGO1-UGA. This clone was then used as a template for in vitro assembly with primers 4–7 to generate UAA, UAG and GCA versions of p2Luc-AGO1-UGA. AGO1-Fluc inserts were subcloned into pDLucBiDi (bidirectional SV40 promoter) (Addgene 219562) with either *Avr*II / *Bam*HI to generate clones with (unfused) StopGo immediately preceding Fluc, or *Avr*II /

*Eco*RI to generate clones without (fused) StopGo immediately preceding Fluc. All clones were verified by Sanger sequencing (Eurofins).

## Cell culture and transfections

HEK293T cells (ATCC) were maintained in DMEM supplemented with 10% FBS, 1 mM L-glutamine and antibiotics. For dual-luciferase assays, cells were transfected with Lipofectamine 2000 reagent (Invitrogen) in 96-well plates using the 1-day protocol in which suspended cells are added directly to the DNA complexes, with 50 ng of each indicated plasmid. The transfecting DNA complexes in each well were incubated with $5 \times 10^4$ cells for 22 h at 37 °C in 5% $CO_2$.

## Dual-luciferase assay

Relative light units were measured on a Veritas Microplate Luminometer with two injectors (Turner Biosystems). Transfected cells were lysed in 20 μl of 1× passive lysis buffer (PLB: Promega) and light emission was measured following injection of 50 μl of either R-luc or F-luc substrate (Dyer et al, 2000).

Recoding efficiencies were determined by calculating relative luciferase activities (F-luc/R-luc) from test constructs and dividing by relative luciferase activities from replicate wells of sense control constructs. Three replicate biological samples were assayed, each with four technical repeats. Statistical significance was determined using a two-tailed, homoscedastic Student's *t* test.

## Peer review information

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

## Acknowledgements

This work was supported by an Advanced Laureate award (IRCLA/2019/74) from the Irish Research Council to JFA.

## Author contributions

**Apoorva N Suresh**: Data curation; Methodology; Writing—review and editing. **John F Atkins**: Formal analysis; Supervision; Funding acquisition; Writing—review and editing. **Gary Loughran**: Conceptualisation; Data curation; Formal analysis; Supervision; Investigation; Methodology; Writing—original draft; Project administration; Writing—review and editing.

## Disclosure and competing interests statement

ANS and JFA declare no competing interests. GL is a shareholder of EIRNABio Ltd.

