## [Peer Review File · The EMBO Journal]

The dual luciferase assay does not support claims of stop codon readthrough on the AGO1 mRNA

Apoorva Suresh, John Atkins, and Gary Loughran

Corresponding author(s): Gary Loughran (g.loughran@ucc.ie)

Review Timeline:

Submission Date:	4th Apr 24
Editorial Decision:	17th Jul 24
Appeal Received:	31st Oct 24
Editorial Decision:	10th Apr 25
Revision Received:	15th Apr 25
Accepted:	13th May 25

Editor: *Cornelius Schneider*

Transaction Report:

Dear Dr. Loughran,

Thank you for submitting your correspondence for consideration by the EMBO Journal. I have shared the manuscript with the authors of the original studies in question (PMID: 31330067 and 38499809) and have sent both your manuscript, as well as the response by Eswarappa and colleagues, for peer review. Please find the response letter and the referee comments included below.

As you can see from the reports, the referees appreciate that the new reporter system established in this manuscript is likely superior to dual luciferase reporter employed in the studies in question and will be an important tool for the field of research. However, the referees also think that Eswarappa and colleagues offer convincing arguments that there is indeed translation readthrough in the AGO1 mRNA. Most importantly, there are independent studies which support the existence of the C-terminally extended AGO1 proteoform. In addition, the referees also think that the differences in 3'UTR sequences employed between your reporter and the reporter used in the studies by Eswarappa and colleagues render both systems incomparable.

We have now extensively discussed this matter within the team and have also engaged in repeated consultations with the referees. After all these considerations we have concluded that the evidence provided here is not sufficient to justify invalidating the two studies by Eswarappa and colleagues and we therefore are not able to offer publication of the correspondence at The EMBO Journal.

That being said, we think that the reporter system described here is interesting and important and that the field of mRNA translation research would benefit from a careful review of reporter systems with all their advantages and caveats. We would be open to such a more general and extended commentary.

Yours sincerely,

Cornelius Schneider

Cornelius Schneider, PhD
Editor
The EMBO Journal
c.schneider@embojournal.org

Referee #1:

In their Correspondence, Apoorva N. Suresh, John F. Atkins and Gary Loughran discuss methodological advances and pitfalls assessing translational readthrough using Luciferase reporter assays. The presented data fail to reproduce published data by two previous studies by Eswarappa SM and colleagues (2019 and 2024). In their rebuttal, Eswarappa and colleagues point at additional evidence for translational readthrough at the AGO-1 gene and list three publications by independent research groups that observed readthrough at AGO-1 (though in other cell lines).

As one of the reviewers of the Singh et al., 2019 manuscript, I would like to offer a few observations. But first, I would like to apologize that my unfamiliarity with readthrough sensors prevented me from observing potential technical issues earlier. As an expert in small RNA biology, I focused on the suggested regulation of readthrough by the microRNA (miRNA) let-7. However, in Singh et al., 2019, the sensor assays were the most convincing data for me and the only data indicating a miRNA-dependent regulation. These data should be re-assessed in the light of new findings by Loughran and colleagues and published data by Ghosh et al. that do not observe robust readthrough at AGO-1 in HEK-203 cells.

While the Eswarappa group cites Ghosh et al 2020 in favor of their observation, Ghosh et al. fail to observe robust readthrough in HEK293 cells (Ghosh et al., EMBO J 2010, Fig. 1D), adding to the potential problem with conclusions on miRNA-regulation of readthrough in Singh et al. 2019 and CRISPR/Cas inhibition in Manjunath et al 2024.

Data in Ghosh et al. 2020 support the data presented by Suresh et al. in this correspondence. There seems to be very little readthrough at AGO-1 in HEK-293 cells. However, Ghosh et al. present a much more elaborate study that identified significant translational readthrough at AGO-1 in other cell lines, especially MDA-MB-231.

In summary, I believe that publishing this correspondence will be of great benefit to a broad audience and further discussions. In

future studies of readthrough at AGO-1, all parties might want to consider changing the tissue culture cell line from HEK-293 to MDA-MB-231 (Ghosh et al., 2020). With the low activity of readthrough at AGO-1 in HEK-293 cells, the attentive reader will have an opportunity to re-evaluate regulatory data presented by the Eswarappa group in 2019 and 2024. The Eswarappa group might want to confirm these observations in more robust cell systems.

Referee #2:

Dear Editor,

You requested that I assess the communication between Loughran and his collaborators and Eswarappa and his colleagues. The disagreement centers on Loughran's group's inability to demonstrate readthrough at the Ago1 gene stop codon using a dual reporter system. I have thoroughly examined both sides' arguments.

While both sides present valid points, there are also inaccuracies on each side. These will be detailed below.

Loughran's group observed an eightfold difference in luciferase activity between unfused and fused plasmids. The only difference between these plasmids was the presence of the SG sequence. Since both plasmids are in-frame for Ago1 and F-luc, they should theoretically produce 100% F-luc activity. This suggests that in the absence of the SG sequence, F-luc activity or synthesis is reduced. However, this eightfold difference disappears when the sense codon (GCA) is replaced with any of the three stop codons. In this case, constructs with or without the SG sequence yield similar results. This introduces a clear bias into the calculation of read-through efficiency. It is unclear whether Loughran's group observed the same phenomenon with the dual reporter system, as they do not address this point.

I disagree with Loughran's interpretation of a background level of luciferase activity. This background activity should also be present with the stop codons, unless it is compensated by a reduced initiation rate or a higher level of mRNA degradation due to nonsense-mediated decay (NMD) in the presence of the stop codon. Investigating the mRNA stability of the constructs would be very informative in this regard.

These observations raise questions about the suitability of using this single reporter system as a tool to study Ago1 read-through.

About the answers from Eswarappa's group

1- Several independent lines of evidence support AGO1 undergoing stop codon readthrough. Among the proposed methods, mass spectrometry data appears to be the most convincing. Particularly, the work by Ghosh et al. provides compelling evidence for the existence of Ago1x (the readthrough form of Ago1). Notably, this study employed cancer cells, which could be a significant factor. Unfortunately, I was unable to find the table EV1 mentioned.

2- The argument regarding the lack of negative controls is questionable. While Loughran et al. included in-frame controls (positive controls), the necessity of negative controls in this specific scenario requires further consideration. Since they were already unable to detect readthrough activity from the Ago1 sequence, it's unclear what information a negative control would provide.

3- Although it is true that RiboSeq data from Bidou et al. detects readthrough in Ago1 this is only clear in presence of G418, a readthrough inducer, so this is not a strong argument in favor of SCR in Ago1. However the various cell lines used can also account for discrepancy.

4- I concur that reporter plasmids can be a source of misinterpretation and require careful application. However, the potential pitfalls outlined here would typically lead to overestimation, not underestimation, of activity. Therefore, I am not persuaded by the argument that these limitations automatically explain the lack of observed readthrough.

5- I did not check the issue with the HindIII site, but clearly differences within the 3' UTR region could account for the absence of readthrough activity. This must be verified carefully.

6- The argument for alternative initiation is not entirely convincing. If this were the case, we would expect to see similar activity in the presence of the stop codon, not just the sense codon. As previously mentioned, investigating mRNA stability and/or initiation rate would be probably more interesting.

In conclusion, discerning between the opposing viewpoints presents a significant challenge. While compelling evidence from mass spectrometry and other research groups supports AGO1 undergoing stop-codon readthrough with a potential physiological role, the SG-F-Luc reporter system employed by Loughran et al. appears to exhibit inconsistencies. Therefore, quantification of readthrough efficiency using this system requires significant caution.

Referee #3:

The authors used a modified version of a stop codon readthrough (SCR) vector containing a StopGo (SG) peptide sequence to separate the two luciferase readouts (Renilla and Firefly). Using transfections, they show a very low level of read-through of AGO1 compared to AQP. They claim that the readthrough reported by Singh et al. 2019 of 20% is not reproduced and is likely a result of the inactivity of their luciferase fused reporter

Singh et al. rebut this claim mainly by indicating the lack of controls, the lack of various independent assays, ignoring supporting evidence from the literature, and a flaw in the construction of the reporter seemingly omitting sequences of the proximal 3'UTR required for the SCR. Furthermore, in their response, Singh et al. provide evidence for absolute luciferase data in their manuscript, indicating that there is no problem in the detection of SCR by their vector system.

Thus, Suresh et al. manuscript suffers from clear deficiencies that cast doubt on the validity of their findings. Singh et al. correctly and significantly pinpoint these deficiencies, and provide further evidence supporting their published observations. Additionally, their observations were independently supported by publications from other labs. Finally, I am mostly concerned about the flaw in the 3'UTR construct made by Suresh et al., which can explain most discrepancies and could have been quickly resolved through direct interaction between the two labs.

** As a service to authors, EMBO Press provides authors with the possibility to transfer a manuscript that one journal cannot offer to publish to another EMBO publication or the open access journal Life Science Alliance launched in partnership between EMBO Press, Rockefeller University Press and Cold Spring Harbor Laboratory Press. The full manuscript and if applicable, reviewers' reports, are automatically sent to the receiving journal to allow for fast handling and a prompt decision on your manuscript. For more details of this service, and to transfer your manuscript please click on Link Not Available. **

Referee #1:

In their Correspondence, Apoorva N. Suresh, John F. Atkins and Gary Loughran discuss methodological advances and pitfalls assessing translational readthrough using Luciferase reporter assays. The presented data fail to reproduce published data by two previous studies by Eswarappa SM and colleagues (2019 and 2024). In their rebuttal, Eswarappa and colleagues point at additional evidence for translational readthrough at the AGO-1 gene and list three publications by independent research groups that observed readthrough at AGO-1 (though in other cell lines).

As one of the reviewers of the Singh et al., 2019 manuscript, I would like to offer a few observations. But first, I would like to apologize that my unfamiliarity with readthrough sensors prevented me from observing potential technical issues earlier. As an expert in small RNA biology, I focused on the suggested regulation of readthrough by the microRNA (miRNA) let-7. However, in Singh et al., 2019, the sensor assays were the most convincing data for me and the only data indicating a miRNA-dependent regulation. These data should be re-assessed in the light of new findings by Loughran and colleagues and published data by Ghosh et al. that do not observe robust readthrough at AGO-1 in HEK-203 cells.

While the Eswarappa group cites Ghosh et al 2020 in favor of their observation, Ghosh et al. fail to observe robust readthrough in HEK293 cells (Ghosh et al., EMBO J 2010, Fig. 1D), adding to the potential problem with conclusions on miRNA-regulation of readthrough in Singh et al. 2019 and CRISPR/Cas inhibition in Manjunath et al 2024.

Data in Ghosh et al. 2020 support the data presented by Suresh et al. in this correspondence. There seems to be very little readthrough at AGO-1 in HEK-293 cells. However, Ghosh et al. present a much more elaborate study that identified significant translational readthrough at AGO-1 in other cell lines, especially MDA-MB-231.

In summary, I believe that publishing this correspondence will be of great benefit to a broad audience and further discussions. In future studies of readthrough at AGO-1, all parties might want to consider changing the tissue culture cell line from HEK-293 to MDA-MB-231 (Ghosh et al., 2020). With the low activity of readthrough at AGO-1 in HEK-293 cells, the attentive reader will have an opportunity to re-evaluate regulatory data presented by the Eswarappa group in 2019 and 2024. The Eswarappa group might want to confirm these observations in more robust cell systems.

We thank referee #1 for their time in carefully reading the relevant publications.

Referee #2:

Dear Editor,

You requested that I assess the communication between Loughran and his collaborators and Eswarappa and his colleagues.

The disagreement centers on Loughran's group's inability to demonstrate

readthrough at the Ago1 gene stop codon using a dual reporter system. I have thoroughly examined both sides' arguments.

We thank referee #2 for their time in carefully reading all of the relevant publications.

While both sides present valid points, there are also inaccuracies on each side. These will be detailed below. Loughran's group observed an eightfold difference in luciferase activity between unfused and fused plasmids.

Yes, but only for the sense in-frame controls.

The only difference between these plasmids was the presence of the SG sequence. Since both plasmids are in-frame for Ago1 and F-luc, they should theoretically produce 100% F-luc activity. This suggests that in the absence of the SG sequence, F-luc activity or synthesis is reduced.

We would be surprised if StopGo had an impact on synthesis (translation) of F-luc, but we agree that it is impacting F-luc activity/folding/or stability.

We would rather say that when F-luc is directly fused to Ago1, F-luc activity is reduced. Otherwise it sounds as though the StopGo sequence itself is somehow stabilising F-luc, whereas it is more likely that it is the StopGo activity (co-translation separation of peptides) that allows F-luc to fold properly.

However, this eightfold difference disappears when the sense codon (GCA) is replaced with any of the three stop codons. In this case, constructs with or without the SG sequence yield similar results. This introduces a clear bias into the calculation of read-through efficiency.

This is exactly our point. The fact that there is no similar 8-fold decrease with the stop codon constructs provides strong evidence that there is little or no readthrough. We should only expect to observe this same 8-fold difference when comparing the stop codon constructs IF there really is readthrough since we know from the sense codon constructs that the activity of F-luc when fused directly to Ago1 is reduced. We appreciate that this was not clear enough in our correspondence and has now been given more emphasis in the accompanying revised version.

'This introduces a clear bias into the calculation of read-through efficiency.' Again this is our main point. Since there is no similar fold decrease with the stop codon constructs without StopGo then there is a clear bias in the calculation of readthrough efficiencies which is compounded by the compromised *fused* sense control. If there really was readthrough, and the stop codon constructs without StopGo reduced 8-fold, then the readthrough efficiency calculations should be equal regardless of whether F-luc is fused to Ago1.

It is unclear whether Loughran's group observed the same phenomenon with the dual reporter system, as they do not address this point.

We did not generate unfused versions of the single mRNA dual luciferase reporters for Ago1 as we have encountered issues with fused reporters (see Khan et al., 2022 and Loughran et al., 2022). Since 2017, unfused single mRNA dual luciferase reporters are our default, and in our opinion are the gold standard for dual luciferase based assays for estimating stop codon readthrough and frameshifting.

I disagree with Loughran's interpretation of a background level of luciferase activity. This background activity should also be present with the stop codons, unless it is compensated by a reduced initiation rate or a higher level of mRNA degradation due to nonsense-mediated decay (NMD) in the presence of the stop codon. Investigating the mRNA stability of the constructs would be very informative in this regard. These observations raise questions about the suitability of using this single reporter system as a tool to study Ago1 read-through.

We are unclear what is meant by single reporter system. We assume two reporters on separate mRNAs is meant (like in Eswarappa and colleagues and our Fig. 2). Although both reporter systems can be useful for measuring readthrough efficiency if reporters are not fused to the test sequence, we prefer to use single mRNA dual reporters as there is no need to control for mRNA levels. Here we used both systems in attempt to understand how Eswarappa and colleagues calculated such high readthrough.

Regarding the background levels, we do not suggest that the F-luc activity of the fused sense control is due to background. We suggest that the very low levels of F-luc activity observed from the stop codon constructs (both fused and unfused) must be derived from the translation of minor aberrant or unexpected mRNAs missing all or part of the Ago1 CDS. Plasmids are notorious for generating aberrant transcripts, either through cryptic promoters or cryptic splicing. We believe that the F-luc activity from the stop codon constructs results from such artefacts for two reasons. Firstly, there is no difference in F-luc activity between different stop codons. It is well established that UGA stop codons are more permissive for readthrough than UAG and UAA. Secondly, as mentioned above, if this F-luc activity were due to bone fide readthrough then we would expect the absolute F-luc activities of the stop codon constructs to have the same fold difference (8 fold) as the sense reporters. We believe that the reason for the misinterpretation of these low background levels of F-luc activity as efficient readthrough is simply due to the calculation of readthrough relative to a compromised sense control (fused).

We feel that these points were not sufficiently clear in our correspondence and have now added a clarifying paragraph.

About the answers from Eswarappa's group

1- Several independent lines of evidence support AGO1 undergoing stop codon readthrough. Among the proposed methods, mass spectrometry data appears to be the most convincing. Particularly, the work by Ghosh et al. provides compelling evidence for the existence of Ago1x (the readthrough form of Ago1). Notably, this study employed cancer cells, which could be a significant factor. Unfortunately, I was unable to find the table EV1 mentioned.

The mass spectrometry result presented by Ghosh may appear convincing. However, it should be borne in mind that mass spectrometry is highly sensitive and

can detect very low abundance peptides. We do not disagree that there may be a very low level of stop codon readthrough into the Ago1 3'UTR. It seems highly likely that all stop codons are read through to some extent at very low and tolerable levels. However, Eswarappa and colleagues report 20% readthrough on Ago1 in HEK293T which is extremely high. We do not believe that their data (this is corroborated by Ghosh's HEK293 data) support such high levels of readthrough.

2- The argument regarding the lack of negative controls is questionable. While Loughran et al. included in-frame controls (positive controls), the necessity of negative controls in this specific scenario requires further consideration. Since they were already unable to detect readthrough activity from the Ago1 sequence, it's unclear what information a negative control would provide.

We agree that negative controls are not necessary if positive in-frame controls are used. It is not advisable to calculate reporter activities relative to a negative control rather than a positive control, as this can result in the unintended consequence of an insignificant number having an outsized effect

Just to add that in our opinion the TAA construct should act as a better negative control than any controls without the Ago1 sequence as it is often the insert (test) sequence that harbours the cryptic splice sites or cryptic promoters.

3- Although it is true that RiboSeq data from Bidou et al, detects readthrough in Ago1 this is only clear in presence of G418, a readthrough inducer, so this is not a strong argument in favor of SCR in Ago1. However the various cell lines used can also account for discrepancy.

Ribosome profiling data are inherently spikey due to lower read densities than RNAseq. We believe that it is very challenging to prove readthrough by ribosome profiling without analysing many datasets – this is especially true when the readthrough ORF is short.

We have carried out our own analysis of publicly available RiboSeq data using aggregated datasets in Ribocrypt (<https://ribocrypt.org>) for Ago1 and AQP4. This data is aggregated from ~3300 ribosome profiling experiments that include ~800 from HEK293 cells. As shown below, there is clear evidence of read density 3' of the AQP4 stop codon that falls off abruptly at the next in-frame stop codon. We do not observe such a read density for Ago1 even though its readthrough efficiency is reported by Eswarappa and colleagues to be 2-3 x higher than for AQP4.

Figure legend - Analysis of publicly available ribosome profiling data for AGO1 and AQP4. In the ORF plot below ribosome profiling tracks, black and white dashes represent stop and ATG codons in each reading frame respectively.

4- I concur that reporter plasmids can be a source of misinterpretation and require careful application. However, the potential pitfalls outlined here would typically lead to overestimation, not underestimation, of activity. Therefore, I am not persuaded by the argument that these limitations automatically explain the lack of observed readthrough.

Agreed

5- I did not check the issue with the HindIII site, but clearly differences within the 3' UTR region could account for the absence of readthrough activity. This must be verified carefully.

This was an error in the writing of the materials and methods and has now been corrected. We used the same 3'UTR region as used by Eswarrappa and colleagues and are happy to provide time stamped sequencing traces if required..

6- The argument for alternative initiation is not entirely convincing. If this were the case, we would expect to see similar activity in the presence of the stop codon, not just the sense codon. As previously mentioned, investigating mRNA stability and/or initiation rate would be probably more interesting.

In conclusion, discerning between the opposing viewpoints presents a significant challenge. While compelling evidence from mass spectrometry and other research groups supports AGO1 undergoing stop-codon readthrough with a potential physiological role, the SG-F-Luc reporter system employed by Loughran et al. appears to exhibit inconsistencies. Therefore, quantification of readthrough efficiency using this system requires significant caution.

We hope that we have clearly outlined above the reasons why we believe that there are no inconsistencies with the SG-F-luc reporter system.

Referee #3:

The authors used a modified version of a stop codon readthrough (SCR) vector containing a StopGo (SG) peptide sequence to separate the two luciferase readouts (Renilla and Firefly). Using transfections, they show a very low level of read-through of AGO1 compared to AQP. They claim that the readthrough reported by Singh et al. 2019 of 20% is not reproduced and is likely a result of the inactivity of their luciferase fused reporter

We thank referee #3 for their time in carefully reading our correspondence and all the relevant publications.

Singh et al. rebut this claim mainly by indicating the lack of controls, the lack of various independent assays, ignoring supporting evidence from the literature, and a flow in the construction of the reporter seemingly omitting sequences of the proximal 3'UTR required for the SCR. Furthermore, in their response, Singh et al. provide evidence for absolute luciferase data in their manuscript, indicating that there is no problem in the detection of SCR by their vector system.

Thus, Suresh et al. manuscript suffers from clear deficiencies that cast doubt on the validity of their findings. Singh et al. correctly and significantly pinpoint these deficiencies, and provide further evidence supporting their published observations. Additionally, their observations were independently supported by publications from other labs. Finally, I am mostly concerned about the flow in the 3'UTR construct made by Suresh et al., which can explain most discrepancies and could have been quickly resolved through direct interaction between the two labs.

We have addressed all of these concerns in our responses to reviewer #2.

Dear Gary,

Thank you very much for submitting your correspondence to the EMBO Journal. We have now finally received and assessed the response by Eswarappa and colleagues and are planning to export and publish both pieces as soon as possible. We only have to ask you for a couple of editorial requirements before we can move on.

- Please remove the figure files from the manuscript file and upload as individual Figure files, figure legends should be placed below the References.
- The author list should be placed below the title, affiliations are missing, corr. author needs to be marked and provided the email address; in eJP: authors' contributions should be labeled for each author in the system
- Please provide a "DISCLOSURE AND COMPETING INTERESTS STATEMENT"
- Please arrange the references in alphabetical order
- Please condense your data into a single figure.

Please reorganize the Source data files to one file/folder for the single main figure.

- Section order should be corrected: Title page - Acknowledgements (if any) - Disclosure and Competing Interests Statement - References - Figure Legend - Table

Would it be possible to re-submit the revised manuscript as soon as possible, possibly early next week as we would very much like to export the correspondence before the Easter break?

With best regards,

Cornelius

Cornelius Schneider, PhD
Editor | The EMBO Journal
c.schneider@embojournal.org

Use the link below to submit your revision:

All editorial and formatting issues were resolved by the authors.

Dear Dr. Loughran,

I am pleased to inform you that your manuscript has been accepted for publication in the EMBO Journal.

Yours sincerely,

Cornelius Schneider, PhD
Editor
The EMBO Journal
c.schneider@embojournal.org
